# Comparative physiological study of sea cucumbers from eastern waters of United States

**Eaint Honey Aung Win, Sinthia Mumu, Nahian Fahim, Kusum Parajuli, Elliott Blumenthal, Rebecca Palu, Ahmed Mustafa**[ID]*

Department of Biological Sciences, Purdue University Fort Wayne, Fort Wayne, IN, United States of America

* mustafaa@pfw.edu

**Data Availability Statement:** All relevant data are within the paper and its Supporting Information files.

## Abstract

Sea cucumbers, belonging to the phylum Echinodermata, are known to possess valuable bioactive compounds that have medicinal properties. In several countries, such as Korea, China, and Japan, they are cultured in the aquaculture industries for food and medicinal purposes. Research has shown that different species of sea cucumbers each possesses unique medicinal values. As a result, we strive towards finding species with better health resilience in aquaculture system to be cultured for nutritional and medicinal purposes. In this paper, we compared the physiological and immunological parameters of three species of sea cucumbers, *Cucumaria frondosa* (*C. frondosa*), *Isostychopus badionotus* (*I. badionotus*), and *Pentacta pygmaea* (*P. Pygmaea*) from the waters of the eastern United States as they have not been studied extensively. Four different cells of sea cucumbers, phagocytic, red spherule, white spherule, and vibratile cells, that contribute to their immunity were counted. *C. frondosa* exhibited the highest concentrations of phagocytic cells, white spherule cells, and vibratile cells, compared to the two other species. Due to its high phagocytic cell concentration, the highest phagocytic capacity was seen in *C. frondosa* although it was not statistically significant. We also observed that *C. frondosa* had the highest total cell count and the highest concentration of coelomic protein among the three species. Lastly, *C. frondosa* possessed the highest lysozyme activity. Taken together, we concluded that *C. frondosa* is the best of the three species compared to be reared in the aquaculture systems for use in the food and biomedicine industries due to its immunological and physiological properties.

## 1. Introduction

Sea cucumbers, also known as Holothurians, are invertebrates that belong to the phylum Echinodermata [1]. These animals have a unique elongated body structure that is supported by a hydrostatic skeleton, and the body muscle is shaped by a fluid-filled cavity called the coelom [2]. In Asian countries, species such as *Stichopus hermanni*, *Thelenota ananas*, *Thelenota anax*, *Holothuria fuccogilva*, and *Actinopyga mauritiana*, are used as a functional food and traditional

**Funding:** The author(s) received no specific funding for this work.

**Competing interests:** The authors have declared that no competing interests exist.

medication [3]. According to previous research, polysaccharides, triterpene glycosides, phenols, and lipids can be isolated from the invertebrate [4, 5]. Several studies have shown that compounds possessing antimicrobial, antioxidant, anticancer, and anti-hyperglycemic activities such as saponins, phenolic compounds, and omega-3, can be extracted from the body wall, tentacles and viscera of sea cucumbers [6]. Due to this, sea cucumbers have the potential to be reared in many aquaculture systems for use in the food and biomedicine industries. Although many species of sea cucumbers are raised in the aquaculture industries of Asian countries, not much research has been conducted on the ability of different sea cucumbers to be ideally cultured in the aquaculture environment of the United States despite their health benefits. As a result, it is important to find the most resilient species of sea cucumbers from the waters of the United States for the aquaculture system.

One of the species, the Orange-footed Sea cucumber or *C. frondosa* can be found in the cold waters of the Atlantic Ocean. *C. frondosa* can be found in the benthic areas of the water, possesses an elongated leathery body and can grow up to 40-50cm in length and 10-15cm in width [6]. This species has been found to have bioactive compounds such as Frondoside A that has the ability to treat pancreatic cancer and breast cancer [7, 8]. As a result, *C. frondosa* is seen as a suitable species to culture industrially for nutraceutical benefits [8]. *Isostychopus badionotus*, or the chocolate chip sea cucumber, is a type of sea cucumber found in the Gulf of Mexico [1, 9]. This species can be found primarily on rocky bottoms between the cracks that are in 2.5m depth of water and they tend to have an average size and weight of 324mm and 628g respectively [10]. Medicinal compounds with anti-coagulant and anti-inflammatory properties have been found in this species, and in several countries, the animal is considered edible for nutritional purposes [11, 12]. *P. pygmaea* is a small, stiff brown sea cucumber found in the Gulf of Mexico [13] that possess bioactive compounds with the potential for SARS-CoV-2 treatment, despite not being known to be edible [14, 15].

In many animals, it has been found that stress-induced changes can suppress the immune response in an animal [16]. For animals in an aquaculture system, stressors that the animals often encounter such as handling stress can reduce the quality of the species [17, 18]. As a result, prior to culturing a species, it is important to evaluate the ability of the sea cucumber to mitigate stress without affecting the overall health and quality of the animal, or its nutritional and medicinal benefits. The immune responses of sea cucumbers depend on the coelomocytes that are in the coelomic fluid [19]. During stressful conditions and pathogenic exposures, these cells participate in maintaining homeostasis and eliminating pathogens [19].

As the sea cucumbers encounter stress or pathogens, the invertebrate undergoes cellular responses to maintain homeostasis and overcome the disease state [20]. In the coelomic fluid of sea cucumbers, there are four major types of coelomocytes (Fig 1): red spherule cells, white spherule cells, phagocytic cells, and vibratile cells [19, 26]. They all have functions relating to antibacterial activity, inflammation, wound healing, encapsulation, graft rejection, and cytotoxic activity in the body of sea cucumbers [21]. Red and white spherule cells were found to secret lipase, peroxidase, and serine proteinase resulting in the breakdown of materials after phagocytosis [22, 23]. Phagocytes contain lysosomal enzymes that ingest and destroy unwanted organisms or particles they encounter [24]. Vibratile cells are highly motile and known to assist in the circulation of the coelomic fluid [25, 26]. The vibratile cells also can degranulate during the clotting process in the animal [25].

Although studies have been conducted on the medicinal and nutritional benefits of sea cucumbers, they have only been focused on species that reside in the East Asian and Middle Eastern countries [2, 3, 27–29]. Due to this, it is important to explore the medicinal properties of sea cucumber species in other parts of the world. One of the areas that were not studied extensively for sea cucumber species and their medicinal benefit is the United States. To

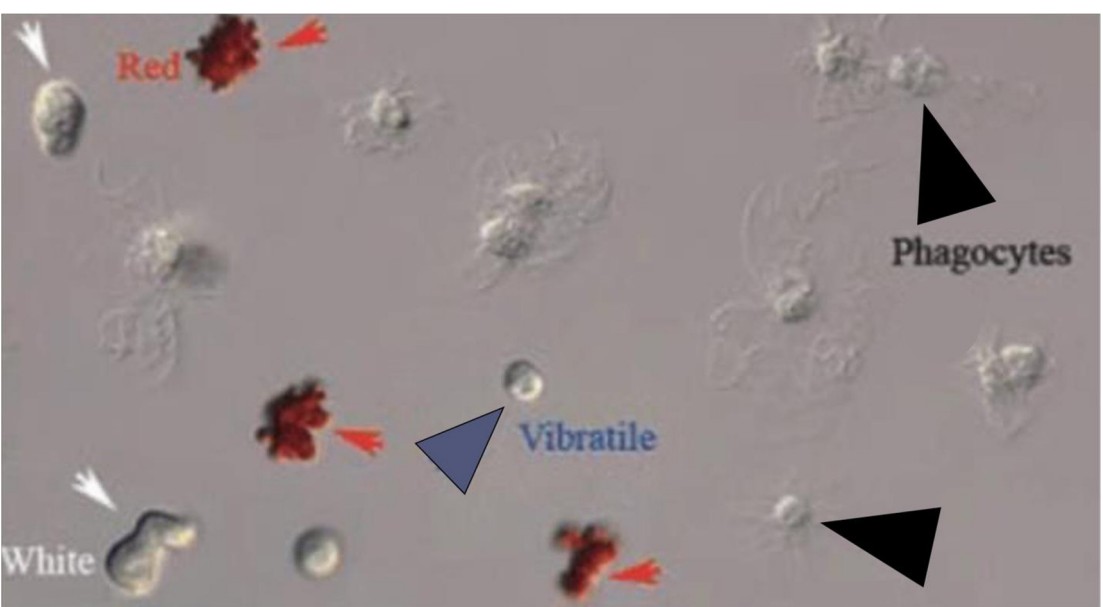

**Fig 1. Types of coelomocytes in echinoderms.** Red arrow = red spherule cells, white arrow = white spherule cells, blue arrow = vibratile cells, black arrow = phagocytic cells [26].

determine the most resilience species best fit for aquaculture, our study evaluates three relevant sea cucumber species from the Eastern United States.

## 2. Material and methods

### 2.1 Collection of coelomic fluid and tissue

Three species of sea cucumbers, *Cucumaria frondosa*, *Isostychopus badionotus*, and *Pentacta pygmaea* were purchased from Gulf of Maine Inc (Pembroke, Maine) and Gulf Specimen Marine Lab (Panacea, Florida) respectively and reared in the aquaculture lab of Purdue University Fort Wayne. Coelomic fluid was collected through midsagittal dissection with a syringe after the animals were euthanized (n = 6 for each species). For some of the species, data collection was completed in technical replicates. The coelomic fluid was immediately kept on ice for analysis of coelomocytes, coelomic protein level, lysozyme activity, and phagocytic capacity.

### 2.2 Total and differential cell count

After obtaining the coelomic fluid samples, 50μl of coelomic fluid was mixed with 50μl of the anticoagulant, Dipotassium Ethylenediamine Tetraacetate (Sequester-Sol, USA). Then, 25μl of the mixture was loaded into a hemocytometer. Four different types of cells, phagocytic cells—red spherule cells, white spherule cells, and vibratile—were counted to find the differential coelomocyte count (DCC) (Fig 1).

Then the differential cell counts were summed up to find the total coelomocytes count (TCC). Top left and right 16 squares of the hemocytometer were counted and averaged. To obtain cells per milliliter, the following equation described was used [30]:

$$\frac{Cells}{mL} = \frac{cell\ count}{number\ of\ counted\ corners} \times dilution\ factor\ (2) \times 10^4$$

## 2.3 Total coelomic protein

Total coelomic fluid protein was measured using a Protein Refractometer (VEEGEE Scientific Inc. Kirkland, WA). Two to three drops of coelomic fluid without anticoagulant were added to the surface of the prism of the calibrated refractometer. Afterward, coelomic protein (g/100ml) was read holding the refractometer under the light.

## 2.4 Lysozyme activity assay

Before initiating the lysozyme assay, a suspension of *Micrococcus lysodeikticus* was made at a concentration of 0.2mg/ml using 0.05M (pH = 6.2) sodium phosphate buffer. 25µl of collected coelomic fluid without anticoagulant was added into a cuvette with 1ml of the bacterial suspension. Then, the absorbance was measured at 1 minute and 5 minutes using a spectrophotometer at 540 nm. Lysozyme activity assay (LAA) was calculated according to the formula [31]:

$$LAA = \frac{(final\ absorbance - initial\ absorbance)}{total\ elapsed\ time\ (minute)}$$

## 2.5 Phagocytic capacity

50µl of coelomic fluid was mixed with 50µl of the anticoagulant. Then, to a double cytoslide microscope slide, 50µl was pipetted into each circle [32]. The slide was incubated for 90–120 minutes at room temperature. After the first incubation, 50µl of formalin-killed bacteria (*Bacillus megaterium*) was added to each circle of the glass slide and was incubated again for 60 minutes at room temperature. When incubation was complete, the slide was washed with phosphate buffer saline (PBS) for 1 minute. Then, it was fixed with methanol for 1 minute, stained with Wright-Giensa stain for 20 seconds, and rinsed with PBS. The slide was air-dried and counted under the microscope. Positive ($\geq$5 engulfed bacteria) and negative phagocytic cells at a location were recorded to find the percent of phagocytic capacity [25].

## 2.6 Statistical analysis

The statistical analysis was conducted using a one-way analysis of variance (ANOVA) in Sigma Plot® 14.5 (Systat Software Inc). To assess statistical significance (p<0.05), pairwise comparisons were performed using a Tukey's HSD test (post ANOVA comparison of multiple means). The data presented in this experiment are illustrated as means ± SEM.

## 3. Results

From the coelomic fluid obtained from each species, four different cell types were counted: phagocytic, red spherule, white spherule, and vibratile (Fig 2). We found phagocytic cell concentrations of 127.83±21.11 x $10^4$ cells/mL, 27.50 ±9.41 x $10^4$ cells/mL, and 56.00±0 x $10^4$ cells/mL for *C. frondosa*, *I. badionotus*, and *P. pygmaea*, respectively. When compared to *I. badionotus* for phagocytic cells concentration, *C. frondosa* had significantly higher values (F = 10.5903572, p<0.05). On the other hand, when the phagocytic cell concentration of *C. frondosa* was compared to *P. pygmaea*, no statistical significance was found (F = 1.23612866, p = 0.30293969). For red spherule cells, there were 14.83±10.52 x $10^4$ cells/mL for *C. frondosa*, 58.50±8.96 x $10^4$ cells/mL for *I. badionotus*, and 13.00±0 x $10^4$ cells/mL for *P. pygmaea*. No significant differences were found among the three species (F = 1.16299833, p = 0.35147122). Regarding white spherule concentration, *C. frondosa*, *I. badionotus*, and *P. pygmaea* had 30.83 ±3.29 x $10^4$ cells/mL, 25.50±8.96 x $10^4$ cells/mL, and 15.00±0 x $10^4$ cells/mL, respectively. No

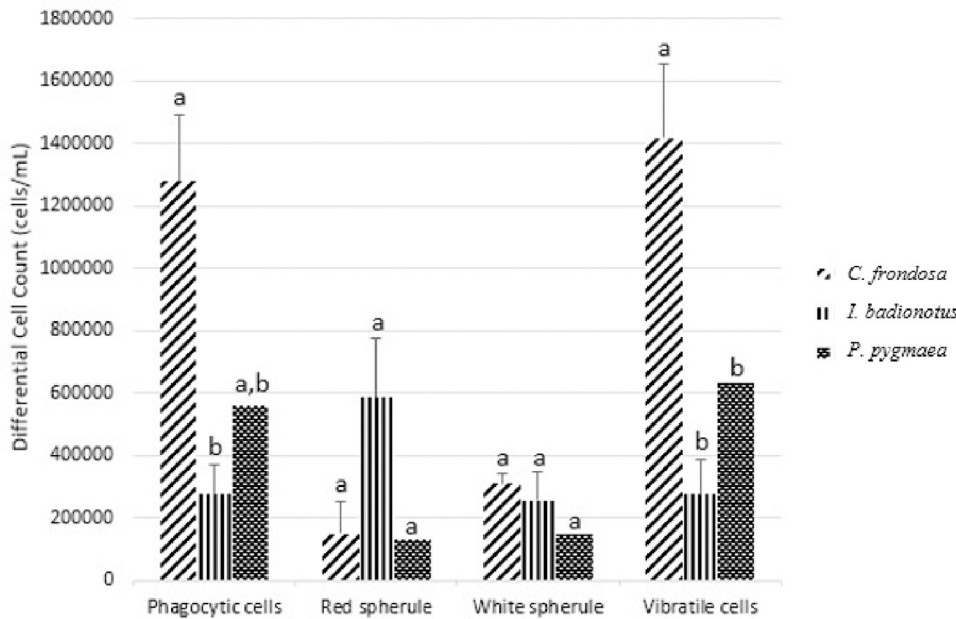

**Fig 2. Differential cell count (cells/mL) of *Cucumaria frondosa* (*C. frondosa*), *Isostychopus badionotus* (*I. badionotus*), and *Pentacta pygmaea* (*P. pygmaea*).** The cell concentrations illustrated are averaged. Different alphabets represent significantly different concentrations among the species for each cell type (p<0.05).

significant differences in white spherule cell concentrations were found among the groups (F = 0.77763541, p = 0.48539954). Lastly, there were 141.67±23.85 x $10^4$ cells/mL, 27.50±11.08 x $10^4$ cells/mL, and 63.00±0 x $10^4$ cells/mL of vibratile cells for *C. frondosa*, *I. badionotus*, and *P. pygmaea* respectively. The concentration of vibratile cells between *C. frondosa* and *I. badionotus* were significantly different (F = 10.9502412, p<0.05). When *C. frondosa* was compared to *P. pygmaea*, no statistical significance was found (F = 3.13936934, p = 0.11971647).

The total cell count was obtained for each type of species (Fig 3). The total cell count concentrations were 315.17±71.65 x $10^4$ cells/mL for *C. frondosa*, 139.00±45.09 x $10^4$ cells/mL for *I. badionotus*, and 147.00±0 x $10^4$ cells/mL for *P. pygmaea*. When the statistical analysis was conducted, the total cell count concentration for *C. frondosa* was significantly different when compared to *I. badionotus* (F = 5.24555283, p<0.05).

Coelomic fluid protein concentration was 1.85±0.09 g/100mL for *C. frondosa*, 2.80±0.07 g/100mL for *I. badionotus*, and 3.58±0.13 g/100mL for *P. pygmaea* (Fig 4). Coelomic protein concentrations were all significantly different from one species to another and *C. frondosa* produced the lowest concentration of coelomic protein compared to *I. badionotus*, and *P. pygmaea* (F = 95.480557, p<0.05).

We also examined the lysozyme activity of the three species to measure the ability of lysozyme present in the coelomic fluid to break down bacterial cell walls (Fig 5). We observed that the activities were 2.60 x $10^{-3}$±5.60 x 10−4 absorbance/minute for *C. frondosa*, 6.50 x $10^{-4}$±7.4 x $10^{-5}$ absorbance/minute for *I. badionotus*, and 1.25 x $10^{-3}$ ±1.50 x $10^{-4}$ absorbance/minute for *P. pygmaea*. After statistical analysis, lysozyme activity of *C. frondosa* was significantly different when compared to *I. badionotus* and *P. pygmaea* (F = 5.68851657, p<0.05).

Lastly, we looked at phagocytic capacity as a measure of immunological status (Fig 6). The percentages of phagocytic capacities were 85.86±4.30%, 76.25±3.33%, and 69.12±4.01% respectively for *C. frondosa*, *I. badionotus*, and *P. pygmaea*. No statistical differences were found between the species (F = 3.86272579, p = 0.05714963).

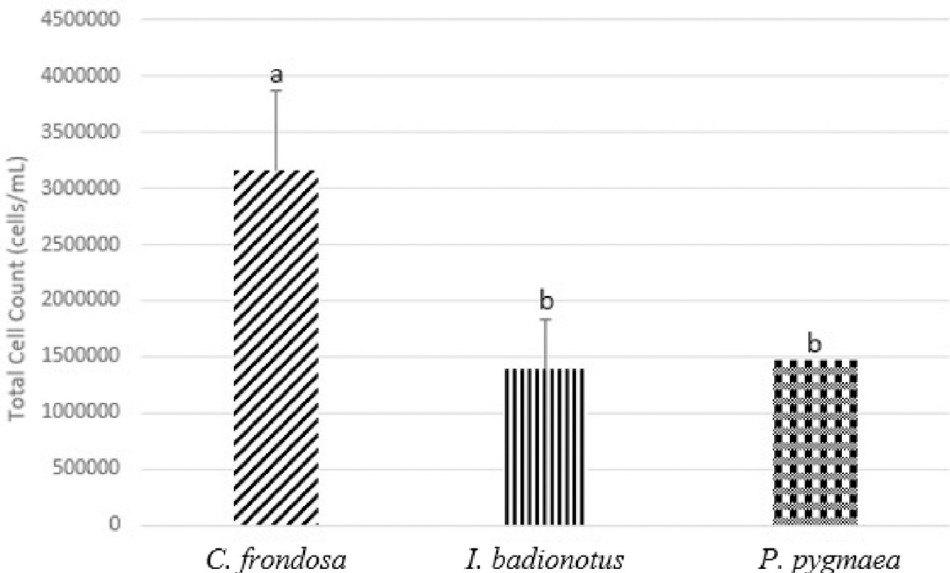

**Fig 3. Total cell count (cells/mL) of *Cucumaria frondosa* (*C. frondosa*), *Isostychopus badionotus* (*I. badionotus*), and *Pentacta pygmaea* (*P. pygmaea*).** The cell counts illustrated are averaged. Different alphabets represent significantly different concentrations of cells among the species (p<0.05).

## 4. Discussion

Based on the results generated in this project, we were able to find the best species with physiological and immunological parameters. We determined this based on the differential cell count concentrations. Overall, we observed that *C. frondosa* had higher cell counts for all cell types counted, as well as the total cell count. This could have been due to the size differences between

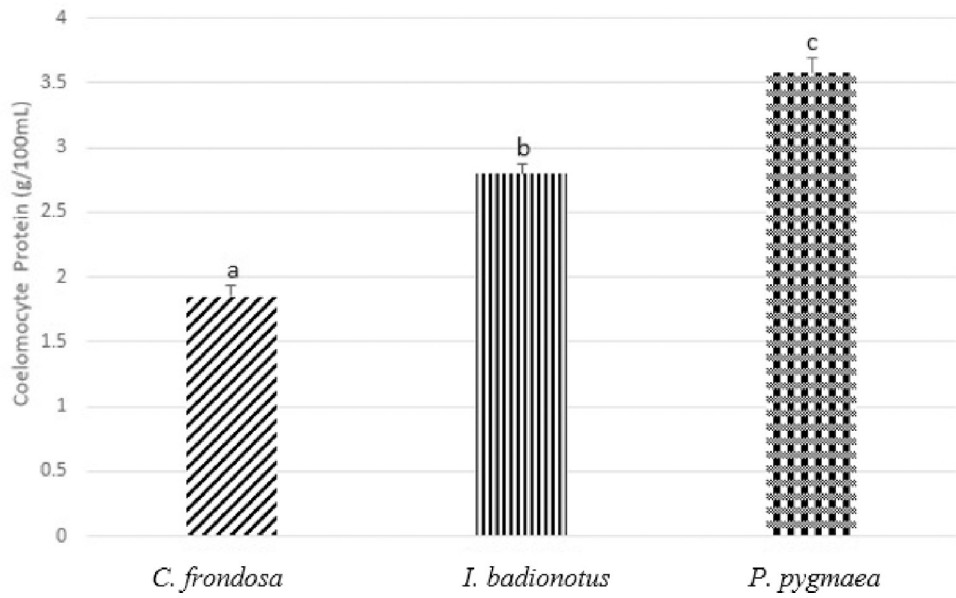

**Fig 4. Coelomic protein (g/100mL) of *Cucumaria frondosa* (*C. frondosa*), *Isostychopus badionotus* (*I. badionotus*), and *Pentacta pygmaea* (*P. pygmaea*).** The concentrations are illustrated in mean +SEM. Different alphabets represent significantly different concentrations of coelomocyte protein among the species (p<0.05).

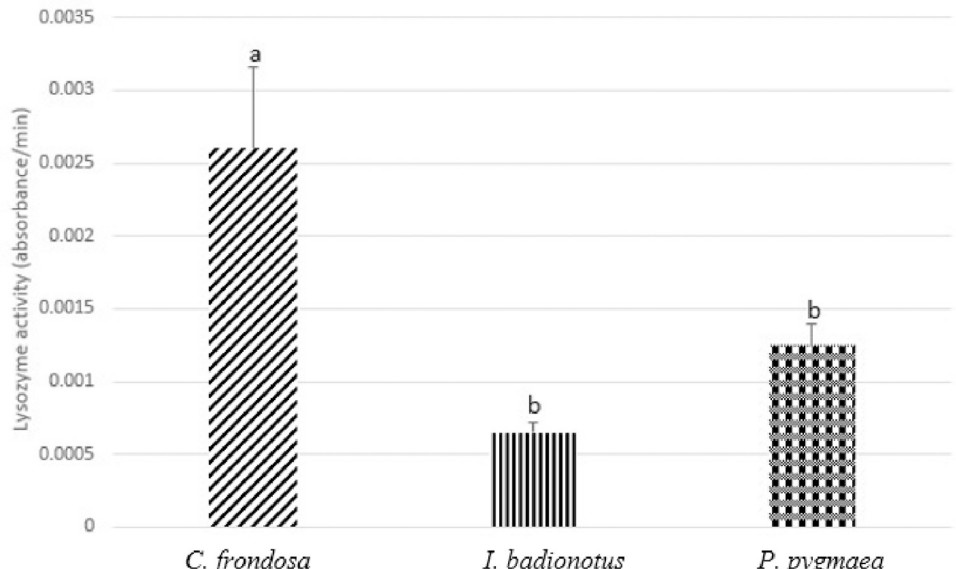

**Fig 5. Lysozyme activity (absorbance/minute) of** *Cucumaria frondosa* **(***C. frondosa***),** *Isostychopus badionotus* **(***I. badionotus***), and** *Pentacta pygmaea* **(***P. pygmaea***).** The activities are illustrated in mean +SEM. Different alphabets represent significantly different lysozyme activity among the species (p<0.05).

the species studied in this research as *C. frondosa* was the biggest species compared to the other two. Other studies supported our results as the authors showed that higher coelomocyte counts and enhanced concentration of phagocytes were seen over time when *C. frondosa* was induced with stress [20, 33]. In our result for differential cell count, we also saw that *C. frondosa* had a significantly high number of phagocytic cells, which could indicate a better immune response. Even though the data analysis showed no significant differences, some phagocytic

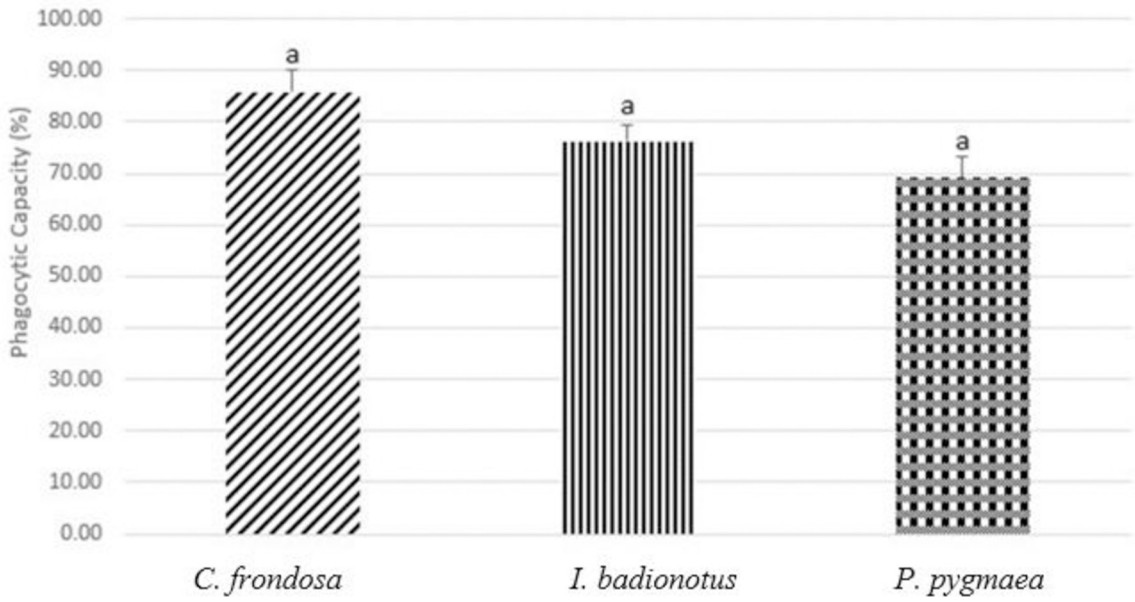

**Fig 6. Phagocytic capacity (%) of** *Cucumaria frondosa* **(***C. frondosa***),** *Isostychopus badionotus* **(***I. badionotus***), and** *Pentacta pygmaea* **(***P. pygmaea***).** The capacities are illustrated in mean +SEM. There were no significant differences among the three species (p>0.05).

capacity was seen in *C. frondosa* as we expected due to higher count of phagocytic cells. Although not all results for differential and total cell counts were significant, we can imply that *C. frondosa* has the highest number of counted immune cells in its coelomic fluid compared to *I. badionotus* and *P. pygmaea*.

During stressful and pathogenic encounters, proteins help maintain homeostasis and fight against pathogens are secreted into the coelomic fluid [34]. We measured the coelomic protein concentrations of the three species and found that *C. frondosa* had the highest concentration of coelomic protein among the three species. In a previous study, it has been shown that sea cucumbers produce an increased number of proteins such as heat shock proteins and lysozymes to maintain homeostasis during stressful situations [35, 36]. In addition to the coelomic protein, we also observed significantly different lysozyme activity when *C. frondosa* was compared to *I. badionotus* and *P. pygmaea*. We found that *C. frondosa* has the highest lysozyme activity and lysozyme activity was also seen in the coelomic fluid of *C. frondosa* in a previous study [37]. It is possible that *I. badionotus* and *P. pygmaea* had less production of coelomic protein compared to *C. frondosa* due to dietary differences and building blocks availabilities in different niches they reside.

## 5. Conclusion

Based on these results, we can conclude that *C. frondosa* has the best immunological and physiological properties among the three species. It is important to find the best sea cucumber species for the aquaculture industry as it needs to be able to handle stress and maintain healthy conditions in order to be cultured effectively for nutritional and medicinal purposes.

## Supporting information

**S1 Data.**
(XLSX)

## Author Contributions

**Conceptualization:** Eaint Honey Aung Win, Sinthia Mumu, Nahian Fahim, Kusum Parajuli, Elliott Blumenthal, Rebecca Palu, Ahmed Mustafa.

**Data curation:** Eaint Honey Aung Win, Nahian Fahim.

**Formal analysis:** Eaint Honey Aung Win.

**Investigation:** Eaint Honey Aung Win, Elliott Blumenthal, Rebecca Palu, Ahmed Mustafa.

**Methodology:** Eaint Honey Aung Win, Sinthia Mumu, Nahian Fahim, Kusum Parajuli, Elliott Blumenthal, Rebecca Palu, Ahmed Mustafa.

**Project administration:** Ahmed Mustafa.

**Resources:** Ahmed Mustafa.

**Supervision:** Ahmed Mustafa.

**Validation:** Eaint Honey Aung Win.

**Writing – original draft:** Eaint Honey Aung Win.

**Writing – review & editing:** Sinthia Mumu, Nahian Fahim, Kusum Parajuli, Elliott Blumenthal, Rebecca Palu, Ahmed Mustafa.

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
