## [Decision Letter · Decision Letter 0]

24 Jul 2023

PONE-D-23-17088Comparative Physiological Study of Sea Cucumbers from Eastern Waters of United StatesPLOS ONE

Dear Dr. Mustafa,

Thank you for submitting your manuscript to PLOS ONE. After careful consideration, we feel that it has merit but does not fully meet PLOS ONE’s publication criteria as it currently stands. Therefore, we invite you to submit a revised version of the manuscript that addresses the points raised during the review process.

We look forward to receiving your revised manuscript.

Kind regards,

Mohammed Fouad El Basuini, Professor

Academic Editor

PLOS ONE

Journal Requirements:

Reviewers' comments:

Reviewer's Responses to Questions

**Comments to the Author**

1. Is the manuscript technically sound, and do the data support the conclusions?

Reviewer #1: Yes

Reviewer #2: Yes

Reviewer #3: Yes

2. Has the statistical analysis been performed appropriately and rigorously? 

Reviewer #1: Yes

Reviewer #2: Yes

Reviewer #3: I Don't Know

3. Have the authors made all data underlying the findings in their manuscript fully available?

Reviewer #1: Yes

Reviewer #2: Yes

Reviewer #3: No

4. Is the manuscript presented in an intelligible fashion and written in standard English?

Reviewer #1: Yes

Reviewer #2: Yes

Reviewer #3: Yes

5. Review Comments to the Author

Reviewer #1: Dear Author,

The article has some interesting observation and information for Comparative Physiological Study of Sea Cucumbers from Eastern Waters of United States. I am informing you that the manuscript has been recommended and acceptance for publication, but also suggest some suggestion to your manuscript. The Introduction to discussion part gives a good informational and original observation context to the study, but moves abruptly without proper linking to a statement of objectives. The results text should focus on which features of the data sets are of particular very interesting whole manuscript.

Authors follow first PLOS ONE general instruction guidelines; carefully check journal formatted of the style.

2. Check the grammar part and plagiarism thoroughly.

3. Authors must incorporate all the corrections which I made in the manuscript.

4. Recent references to add me for your future research and outcome of your manuscript.

5. Attention may please be given to the corrections or modifications indicated in the enclosed manuscript while typing the revised manuscript.

6. Whole manuscript thoroughly checks plagiarism part very important and improve the research.

7. Line No. 20, introduction I. badionotus should be Isostychopus badionotus.

8. Most of the place mentioned reference in number, but some other place has been cited author and year? i.e. Xu et al. (2018), Acosta et al. (2021), Yeh et al. (2004), Mumu & Mustafa (2022), Jobson et al. (2021),

9. Figure number 1.1. not available anywhere in the MS, what is the meaning of the figure 2.1, 2.2, 2.3 etc and also figure caption always should be bottom of the figure.

Best Wishes

Thanking you,

Thanking you,

Reviewer #2: The research paper was well written and explained properly.

In the entire research paper et al must to change to italics.

Graps present very good manner.

Methodology present written and explained properly.

Reviewer #3: Overall, the science is good, and the results are conclusive. However, I think the authors need to develop their narrative and discussion. Generally, statements in the introduction and abstract are broad without development or many connecting ideas. The discussion doesn’t provide any insight into implications or steps forward. It is just reiterative of the results. The authors must focus on concision and ask themselves, “Is this informative or just taking up space? What does the reader need to know to understand the importance of my research? What does the reader need to know in order understand why I did the research with these methods?”

That said, I do think that after revision, this paper may be fit for publication in PLOS ONE.

The real issues that limit it from publication in any journal are in reference to statistical tests. Statistical tests were not described in the methods, which makes p-values listed in the results meaningless. Unfortunately, this is accompanied by the lack of a public database, so I cannot even conduct my own statistical tests to validate their results as a reviewer. Luckily, this should be an easy fix for the authors during revisions. More details are provided below.

There are also some issues with the first figure that must be fixed prior to publication.

I did not edit grammatical errors, and instead focused on the structure of the paper and development of ideas. Many issues were gleaned over, and in my comments, I use examples to illustrate my point. My comments are not an exhaustive list of things that should be revised prior to publication. More specific comments are below.

Good luck with revisions, and I look forward to reading it again in a few months.

Abstract:

Sentences 2 and 3 would benefits from an explicit connection between ideas. How is their use in medicine linked to the need to better understand cucumbers' physiological and immunological properties?

I also wonder why, if Korea, China, and Japan are the countries culturing cucumbers, what is the point of studying American species? Similarly, why did you count these cells? Can you briefly state their function and why they are important?

The conclusion is good, just need to be more explicit with how your ideas are connected and their implications.

Include your research question. It is currently stated as the last sentence of the manuscript.

I’m sure your study is valuable; I just think you need to explicitly validate why you completed your study. Otherwise, people won’t understand the point of your paper and opt to read other articles. Each idea in your abstract is distinct, and without more context, I find it difficult to understand how all your ideas are connected. Your introduction paragraphs do a better job at outlining the relevance of your research and comparative studies. Your abstract would benefit from the incorporation of a few of these introductory ideas.

Keywords:

The keywords are more phrases than words. Perhaps opt for something along the lines of: “sea cucumbers, Cucumaria frondosa, Isostychopus badionotus, Pentacta pygmaea, physiology, immunology, aquaculture”

Introduction:

One thing that should be resolved is the lack of detail pertaining to the medicinal and nutritional benefits of consuming sea cucumbers in the first two paragraphs. See the following sentence: “Several studies have shown that these compounds have anti-cancer, antiinflammatory, anti-microbial, anti-angiogenic, anti-hypertension, anti-hyperglycemic, antioxidant, and immunomodulatory activities [6].” This sentence (from your first paragraph) does a nice job of introducing some benefits. However, as you continue through your introduction, you don’t expand on these at all. Do species differ in their benefits? How is this determined? If it’s determined by the cells you counted, explicitly state that. If you are purely evaluating the physiology and immunology of the cucumbers to select the most resilient species for aquaculture, independent from human health benefits, explicitly state that. It feels like there is a lot of missing information that would benefit the reader before diving into your study.

Perhaps I am feeling this way due to a general lack of concision that makes the paper feel ‘fluffy’.

I acknowledge that you do explicitly state potential benefits sometimes. For example: “Although this species is not known to be edible, it has been found to have bioactive compounds with the potential to treat SARS-CoV-2 [15].” However, statements like this are washed out by generally uninformative broad statements: “Due to its medicinal and nutritional benefits, C. frondosa is seen as a species to further culture in the aquaculture industry [8].”

Take the following three sentences:

“The last species in this study is P. pygmaea and it resides in the water of the Gulf of Mexico [13]. This species is brown in color and is considered to be small in size (4-6cm) with ossicles in its body wall that makes it stiffer than other sea cucumbers [14]. Although this species is not known to be edible, it has been found to have bioactive compounds with the potential to treat SARS-CoV-2 [15].”

This could easily be reduced to the following:

P. pygmaea is a small, stiff brown sea cucumber found in the Gulf of Mexico [13] that possesses bioactive compounds with potential for SARS-CoV-2 treatment, despite not being known to be edible [14,15].

If there is limited information or evidence on the medicinal and nutritional benefits of these cucumbers, that’s fine. Just be explicit about the knowledge gaps.

The last two paragraphs in your introduction do a much better job than the first two. This is where your research question is finally outlined: Which cucumber has the highest tolerance to stress and is best fit for aquaculture? State this research objective explicitly both in the abstract and the introduction. It gets lost in the manuscript’s current structure. Then follow the research question with describing the information the reader needs to answer this question. (This concept should also be incorporated into your abstract, which currently misses the mark.)

The last sentence would benefit from some development:

“In this experiment, we study the immunological and physiological properties of coelomic fluid from three sea cucumber species collected from the waters of the Eastern United States: Cucumaria frondosa (C. frondosa), Isostychopus badionotus (I. badionotus), and Pentacta pygmaea (P. Pygmaea). The physiological parameters measured in this study will tell us about the overall health of the sea cucumbers.”

Are you measuring health or are you measuring resilience? If one or the other, explain why in the previous paragraphs. Why only state physiological parameters? Again, why are you testing resilience? Maybe a sentence like this would be more effective: “Our study evaluates three relevant sea cucumber species from the Eastern United States, to infer the most resilient species best fit for aquaculture.”

Materials and Methods:

For this section, I do not have the expertise to evaluate the preparation methodology in depth. However, the replication of n=6 seems fine and the preparation of samples on ice immediately following euthanasia is a generally good practice. Results discussed later seem reliable pending inclusion of statistical methods.

“The tissues (Body Wall, Viscera, and Tentacle) obtained from these invertebrates were kept at -80◦C after rinsing for future experiments.” I don’t think this needs to be stated unless the samples are formally reposited into a collection with collection numbers.

Statistical tests are not described in the materials and methods. This must be included, or the p values listed in the results have no meaning.

Results

No information on tests used for statistical comparison, accompanied by the (P>0.05 and P<0.05) (p should be lower-case) makes me unable to evaluate the results. If p is greater than 0.05, report the exact digit. Also, include the test-statistic value that is derived alongside the p-value. This helps readers evaluate the strength of your results and conclusions. Until this issue is resolved, the manuscript is not fit for publication.

I will more critically evaluate the results upon revision.

Discussion

I hold the similar qualms with the discussion as I do with the introduction. Language is generally vague without much in-depth insight. You only include a couple extra studies with similar results. I would really love to see some conceptual development to better understand why C. frondosa had better immunological and physiological properties. Following development of these ideas, could you provide a framework for how this could be implemented in aquacultural practices?

There is also no discussion about the other two species with comparatively worse performance. How does this compare to the general literature? Should we be concerned about these species’ survival amidst climate change? Or are they just not good candidates for aquaculture?

Let me be clear, I am not asking for tangents or speculation beyond what your results present. However, it would be nice to see further development of ideas and implications. The discussion section as it stands is more of a results section with evidence, and not a real discussion about the concepts evaluated.

Figures

There are no figure numbers associated with your figures.

Cell identification figure: the arrows are not always explicit in what they are pointing to. If possible, it would be nice to move the arrows so that they are directly pointing to the cells. It also seems like this figure has been cropped from a larger panel. There is a white line on the right indicating this. There is also the letter A at the bottom left corner, but it is not a multi-panel figure. The scale bar is not defined in the figure caption either.

The 6 bar plots are good. I do think that they would be more effective as a single figure panel consisting of 6 subpanels. E.g., Figure2A-F instead of Figures 2-7. This would be easier for the reader to get a big picture of your results, but also easier for the typesetter, so that all the figures aren’t mashed into a relatively short paper.

Data availability

I would suggest uploading generated data as supplemental material, instead of relying on readers to email you if interested. Truly open science makes all data accessible immediately.

6. PLOS authors have the option to publish the peer review history of their article (what does this mean?). If published, this will include your full peer review and any attached files.

Reviewer #1: No

Reviewer #2: No

Reviewer #3: **Yes: **Colin Jeffrey Anthony

<quillbot-extension-portal></quillbot-extension-portal>

---

## [Author Response · Author response to Decision Letter 0]

28 Jul 2023

A separate response file with details has been uploaded as "Responses to Reviewers'

July 28, 2023

To:

Professor Mohammed Fouad El Basuini

Academic Editor 

PLOS ONE

Manuscript Number: PONE-D-23-17088

Subject: Responses to the reviewers’ comments for Manuscript: PONE-D-23-17088

Dear Editor,

Greetings.

Please review our responses below to the comments made by the reviewers. We believe that upon reviewing our responses, you will be kind enough to accept our manuscript for publication in PLOS ONE.

Reviewers Responses

Reviewer 1

We would like to thank Reviewer 1 for recommending our following manuscript for publication. 

PLOS ONE general instruction and guidelines were checked and carefully applied to the manuscript.

Grammar and plagiarism was double checked thoroughly.

Corrections the reviewer provided was carefully considered and incorporated. 

Attention was given to the corrections or modifications in the enclosed manuscript.

Species name on Line No. 20 was corrected to Isostychopus badionotus

References have been cited correctly and uniformly. 

Figure numbers were corrected to Figure 1, 2, 3, 4, 5, and 6.

Reviewer 2

We would like to appreciate Reviewer 2 for the kind words and strong recommendation for acceptance of this article, and we italicized et al in the paper. 

Reviewer 3

We are thankful of Reviewer 3’s contribution to helping us improve the manuscript for publication. We took careful consideration of the comments and in cooperated them into the paper. 

The abstract was modified using the comments given by the reviewer we also added the research question in this section and clarified the importance of the species and their function. 

The keywords recommended were added into the manuscript. 

To better the flow of the introduction, additional information and languages were added into the introduction. We also made sure that we explicitly state the research question and purpose of the research. We appreciate the comments very much. 

Comments were in cooperated into the manuscript and the method used to do statistic was added. 

p was changed to lower case in the results and F values/ test-statistic values were added if p was greater than 0.05.

After reviewing the comments, the discussion was modified so the language seems more precise. We also added some conceptual developments to address why one of the species was better than the other

Figure numbers were changed and checked according to the reviewer’s comments. 

Raw Data will be submitted together with the revised manuscript.

We believe that this revised manuscript if in good shape and in right format for publication in PLOS ONE.

We would like to thank the reviewer and the editor again for their kind consideration and contribution. 

Thank you,

Ahmed Mustafa, Ph.D.

Professor, Department of Biological Sciences

Director, Life Sciences Resource Center

Purdue University Fort Wayne

2101 E. Coliseum Blvd., Fort Wayne, IN 46805, USA

Email: mustafaa@pfw.edu; amustafa@purdue.edu

Website: https://users.pfw.edu/mustafaa/

---

## [Decision Letter · Decision Letter 1]

15 Aug 2023

PONE-D-23-17088R1Comparative Physiological Study of Sea Cucumbers from Eastern Waters of United StatesPLOS ONE

Dear Dr. Mustafa,

Thank you for submitting your manuscript to PLOS ONE. After careful consideration, we feel that it has merit but does not fully meet PLOS ONE’s publication criteria as it currently stands. Therefore, we invite you to submit a revised version of the manuscript that addresses the points raised during the review process.

We look forward to receiving your revised manuscript.

Kind regards,

Mohammed Fouad El Basuini, Professor

Academic Editor

PLOS ONE

Journal Requirements:

Reviewers' comments:

Reviewer's Responses to Questions

**Comments to the Author**

1. If the authors have adequately addressed your comments raised in a previous round of review and you feel that this manuscript is now acceptable for publication, you may indicate that here to bypass the “Comments to the Author” section, enter your conflict of interest statement in the “Confidential to Editor” section, and submit your "Accept" recommendation.

Reviewer #3: All comments have been addressed

Reviewer #4: All comments have been addressed

2. Is the manuscript technically sound, and do the data support the conclusions?

Reviewer #3: Yes

Reviewer #4: Yes

3. Has the statistical analysis been performed appropriately and rigorously? 

Reviewer #3: Yes

Reviewer #4: Yes

4. Have the authors made all data underlying the findings in their manuscript fully available?

Reviewer #3: Yes

Reviewer #4: Yes

5. Is the manuscript presented in an intelligible fashion and written in standard English?

Reviewer #3: Yes

Reviewer #4: Yes

6. Review Comments to the Author

Reviewer #3: The authors have responded to all of my previous comments. The revisions were not overly thorough, but they did satisfy all previous requests. I can confirm that the science seems robust and reproducible. I approve this article for publication after the next round of minor revisions. I do not need to see this article again prior to publication.

Specific recommendations (allowed to ignore if the authors disagree):

- p values aren't necessary in the abstract

- remove the word "and" in the keywords

- change "physiological" to physiology in keywords

- "According to research" doesn't add anything to your narrative (e.g. line 101). Either specify the authors or remove the phrase

- line 143. "N" is used for population size, while "n" is used for sample size. Change N to n.

-line 154. remove the word described or say "As described in Yeh et al. [30], the following equation was used to calculate cells per milliliter:"

-In reference to the results section, I believe F typically comes before p

Reviewer #4: Well-planned and organized manuscript. I have very minor comments:

Keywords should be revised.

Line 154: remove "as described in"

7. PLOS authors have the option to publish the peer review history of their article (what does this mean?). If published, this will include your full peer review and any attached files.

Reviewer #3: **Yes: **Colin J Anthony

Reviewer #4: No

---

## [Author Response · Author response to Decision Letter 1]

29 Aug 2023

Responses to reviewers

Reviewers 1 and 2

•We thank reviewers 1 and 2 for their positive acceptance. 

Reviewer 3

Comments/suggestions/recommendations:

•p values aren't necessary in the abstract

•remove the word "and" in the keywords

•change "physiological" to physiology in keywords

•"According to research" doesn't add anything to your narrative (e.g. line 101). Either specify the authors or remove the phrase

line 143. "N" is used for population size, while "n" is used for sample size. Change N to n.

•line 154. remove the word described or say "As described in Yeh et al. [30], the following equation was used to calculate cells per milliliter:"

•In reference to the results section, I believe F typically comes before p

Our response:

•We would like to thank Reviewer 3 for the comments, and we have changed the following recommended comments for the paper.

Reviewer 4

Comments/suggestions/recommendations:

•Line 154: remove "as described in"

Our response:

•We appreciate the kind words from Reviewer 4 and we removed the phrase mentioned by Reviewer 4.

We hope to get the acceptance at your earliest convenience.

Thanks.

---

## [Editor Report · Decision Letter 2]

13 Oct 2023

Comparative Physiological Study of Sea Cucumbers from Eastern Waters of United States

PONE-D-23-17088R2

Dear Dr. Mustafa,

We’re pleased to inform you that your manuscript has been judged scientifically suitable for publication and will be formally accepted for publication once it meets all outstanding technical requirements.

Kind regards,

Mohammed Fouad El Basuini, Professor

Academic Editor

PLOS ONE
---

## [Editor Report · Acceptance letter]

20 Oct 2023

PONE-D-23-17088R2 

Comparative Physiological Study of Sea Cucumbers from Eastern Waters of United States 

Dear Dr. Mustafa:

I'm pleased to inform you that your manuscript has been deemed suitable for publication in PLOS ONE. Congratulations! Your manuscript is now with our production department. 

Kind regards, 

on behalf of

Dr Mohammed Fouad El Basuini 

Academic Editor

PLOS ONE